# Biocompatibility of Small-Diameter Vascular Grafts in Different Modes of RGD Modification

**DOI:** 10.3390/polym11010174

**Published:** 2019-01-18

**Authors:** Larisa V. Antonova, Vladimir N. Silnikov, Victoria V. Sevostyanova, Arseniy E. Yuzhalin, Lyudmila S. Koroleva, Elena A. Velikanova, Andrey V. Mironov, Tatyana S. Godovikova, Anton G. Kutikhin, Tatiana V. Glushkova, Inna Yu. Serpokrylova, Evgeniya A. Senokosova, Vera G. Matveeva, Mariam Yu. Khanova, Tatiana N. Akentyeva, Evgeniya O. Krivkina, Yulia A. Kudryavtseva, Leonid S. Barbarash

**Affiliations:** 1Research Institute for Complex Issues of Cardiovascular Diseases, Kemerovo 650002, Russia; antonova.la@mail.ru (L.V.A.); sevostv@gmail.com (V.V.S.); veliea@cardio.kem.ru (E.A.V.); a.mir.80@mail.ru (A.V.M.); antonkutikhin@gmail.com (A.G.K.); bio.tvg@mail.ru (T.V.G.); sergeewa.ew@yandex.ru (E.A.S.); matveeva_vg@mail.ru (V.G.M.); Khanovam@gmail.com (M.Y.K.); t.akentyeva@mail.ru (T.N.A.); leonora92@mail.ru (E.O.K.); jackie1970@mail.ru (Y.A.K.); reception@kemcardio.ru (L.S.B.); 2Institute of Chemical Biology and Fundamental Medicine of the Siberian Branch of the Russian Academy of Sciences, Novosibirsk 630090, Russia; silnik@niboch.nsc.ru (V.N.S.); koroleva@niboch.nsc.ru (L.S.K.); godov@niboch.nsc.ru (T.S.G.); inna_serpokrylova@biosset.com (I.Y.S.)

**Keywords:** RGD peptides, vascular grafts, arterial replacement, tissue engineering, endothelialisation

## Abstract

Modification with Arg-Gly-Asp (RGD) peptides is a promising approach to improve biocompatibility of small-calibre vascular grafts but it is unknown how different RGD sequence composition impacts graft performance. Here we manufactured 1.5 mm poly(3-hydroxybutyrate-*co*-3-hydroxyvalerate)/poly(ε-caprolactone) grafts modified by distinct linear or cyclic RGD peptides immobilized by short or long amine linker arms. Modified vascular prostheses were tested in vitro to assess their mechanical properties, hemocompatibility, thrombogenicity and endothelialisation. We also implanted these grafts into rat abdominal aortas with the following histological examination at 1 and 3 months to evaluate their primary patency, cellular composition and detect possible calcification. Our results demonstrated that all modes of RGD modification reduce ultimate tensile strength of the grafts. Modification of prostheses does not cause haemolysis upon the contact with modified grafts, yet all the RGD-treated grafts display a tendency to promote platelet aggregation in comparison with unmodified counterparts. In vivo findings identify that cyclic Arg-Gly-Asp-Phe-Lys peptide in combination with trioxa-1,13-tridecanediamine linker group substantially improve graft biocompatibility. To conclude, here we for the first time compared synthetic small-diameter vascular prostheses with different modes of RGD modification. We suggest our graft modification regimen as enhancing graft performance and thus recommend it for future use in tissue engineering.

## 1. Introduction

High prevalence of cardiovascular disease creates a demand for small-diameter vessel grafts used as a bypass for surgical treatment of vascular pathology. However, there are no commercially available vascular prostheses of small diameter (<3 mm), as previous attempts to develop such implants using materials like expanded polytetrafluoroethylene (ePTFE) or polyethylene terephthalate failed due to unacceptably high rates of thrombosis and intimal hyperplasia [1]. As such, more functional and reliable solutions for small-calibre applications are needed.

A promising alternative is a tissue-engineered, biodegradable graft which can be gradually replaced by host endothelial cells thus forming a de novo blood vessel [2]. This can be achieved through modification of luminal surface by growth factors and/or cell adhesion peptides in order to recruit endothelial progenitor cells and stimulate their attachment [1,3]. Accumulating experimental evidence suggests that scaffold backbone supplemented by bioactive molecules is beneficial over standard unmodified grafts [4,5,6].

Arg-Gly-Asp (RGD) is a common cell adhesion sequence displayed by multiple extracellular matrix proteins, to name a few, fibronectin, laminin and fibrinogen [7]. RGD motif acts as a ligand for integrins, a major class of receptors in control of cell adhesion and proliferation [8]. In comparison with bulky protein molecules, RGD-containing peptides are chemically more stable and easily immobilize on polymer surface owing to a simpler structure [3]. Besides cell adhesion enhancement [9], they also favour proliferation as demonstrated by studies on different cell lines cultured on RGD-treated poly(ε-caprolactone) (PCL) films [10]. An interesting question is whether RGD peptide composition may affect biocompatibility of modified implants. Various sequences of RGD peptides have been tested so far, yet there has been no comprehensive comparative study to reveal advantages of one over another [11]. In addition, RGD-containing peptide immobilization requires pre-treatment of the implant by linker molecules and the choice of such linker can also substantially affect graft performance. For example, some studies implicated lengthier linker groups as more effective for fibroblast adhesion and proliferation in vitro [12,13].

Thus, it is currently unknown how different modes of RGD modification influence vascular graft functionality. Previously we showed promising results for 2-mm-diameter polymer RGD-modified implants in vivo [14]; nonetheless, the variety of RGD molecules and linkers for their immobilization demand more in-depth investigations in order to develop the optimal graft formula.

Here, we modified the surface of biodegradable vascular grafts with various RGD peptides using two linkers differing in their length. We show that modification of poly(3-hydroxybutyrate-*co*-3-hydroxyvalerate)/poly(ε-caprolactone) (PHBV/PCL) prostheses using cyclic Arg-Gly-Asp-Phe-Lys peptide combined with 4,7,10-trioxa-1,13-tridecanediamine greatly improves adhesion and viability of endothelial cells whilst not adversely affecting their hemocompatibility. Implantation of these grafts into rat abdominal aortas demonstrated their complete primary patency after 3 months along with formation of an endothelial monolayer.

## 2. Materials and Methods

### 2.1. Ethics

The study was approved by the local ethical committee of the Research Institute for Complex Issues of Cardiovascular Diseases. All subjects involved in the study signed an informed consent after receiving a full explanation of study.

### 2.2. Graft Fabrication

Biodegradable vascular grafts with a luminal diameter of 1.5 mm and 4 mm were electrospun using a Nanon-01A instrument (MECC, Fukuoka, Japan). The 4-mm-diameter prostheses were used for in vitro assays, whilst 1.5-mm-diameter grafts were employed for in vivo implantation. 4 mm diameter grafts were used in vitro because they are much easier to handle and because they have a larger area enabling studies of cell adhesion and proliferation. The polymer blend used for prosthesis fabrication contained 5% *w*/*v* PHBV (Sigma, St. Louis, MO, USA) and 10% *w*/*v* PCL (Sigma) dissolved in trichloromethane. Electrospinning parameters were as follows: voltage 20 kV, feed rate 0.5 mL/h, collector rotation speed 200 rpm and 150 mm tip-to-collector distance. A metal pin with a diameter of 1.5 mm or 4 mm was used as a collector.

### 2.3. Graft Modification with Amine Linkers

Amines purchased from Sigma were used as linker arms for RGD-containing peptides. We used 1,6-hexamethylenediamine and 4,7,10-trioxa-1,13-tridecanediaminemine hereinafter labelled Amine1 and Amine2. To optimize the aminolysis time, prostheses were immersed in a 10% solution of either of the amines prepared in a mixture of isopropanol-water (1:1) and incubated at 37 °C for 10, 30 or 60 min. Samples were then washed with double distilled water and air-dried with the following incubation with 1% ethanol solution of ninhydrin (Sigma) in the presence of 0.05% ascorbic acid (Sigma) at 80 °C for 30 min. Grafts were again washed with ethanol, air-dried and dissolved in 500 μL chloroform. Finally, 500 μL isopropanol was added to the resulting solution and the optical density was measured at a wavelength of 568 nm to determine the number of amino groups.

### 2.4. Immobilization of RGD-Containing Peptides on Graft Luminal Surface

Prostheses were modified by the following RGD-containing peptides: linear peptide RGDK (NanoTech-S, Novosibirsk, Russia) hereinafter labelled Pep1; linear peptide AhRGD (NanoTech-S, Novosibirsk, Russia) hereinafter labelled Pep2; cyclic peptide c[RGDFK] (NanoTech-S, Novosibirsk Russia) hereinafter labelled Pep3 (see Figure 1A for complete peptide sequence). Graft aminolysis lasted 60 min for Amine1 and 30 min for Amine2 based on the abovementioned experiments. Prostheses were successively washed in a mixture of isopropanol-water (1:1), double distilled water, 0.1% Triton X-100 and double distilled water. Grafts were next incubated in 2% aqueous glutaraldehyde (Sigma) at room temperature (RT) for 3 h, washed with double distilled water and further incubated at RT for 4 h with 0.2 mg/mL of Pep1, Pep2 or Pep3 prepared in 50 mM carbonate buffer (pH = 8.5) containing 2.5 mM sodium cyanoborohydride. After peptide attachment, grafts were sequentially washed with 0.1% Triton X-100 and double distilled water.

Eventually, 6 different modifications of biodegradable vascular grafts were obtained: Amine1Pep1, Amine1Pep2, Amine1Pep3, Amine2Pep1, Amine2Pep2 and Amine2Pep3.

### 2.5. Arginine Positivity Test (Sakaguchi Test)

Prostheses previously modified with RGD-containing peptides (1 cm^2^) were placed in 0.1% acetone solution of 8-hydroxyquinoline for 1 min. Samples were then air-dried and immersed in a 0.2% *v*/*v* bromine solution prepared in 0.5M NaOH. Samples were next incubated for 12 h at RT. Orange staining of samples indicated the presence of the arginine guanidino group.

### 2.6. Tensile Testing

To evaluate the mechanical properties of prostheses, uniaxial tension test was performed. Grafts were cut in the longitudinal axis using a custom-shaped knife in the Zwick/Roell cutting press. Segments of human internal mammary artery (length = 10 mm) excised during coronary artery bypass graft surgery were used for control purposes.

Tests were performed on the universal testing machine series Z (Zwick/Roell) using a sensor with a nominal force of 50 N with a limit of permissible error of ±1% and crosshead speed of 50 mm/min. We evaluated ultimate tensile strength, elongation at break and Young’s modulus determined in the range of physiological pressure (80–120 mmHg). Prior to tensile testing, graft samples were not sterilized.

### 2.7. Haemolysis Testing

To assess graft-induced haemolysis, the whole peripheral blood withdrawn from healthy volunteers was mixed with 3.8% sodium citrate at a ratio of 1:9 (citrate:blood). 25 cm^2^ prostheses (*n* = 5 samples per group) were placed in buckets with the following addition of 10 mL saline. Buckets were placed at 37 °C for 2 h and 200 mL citrated blood was added to each bucket before being incubated at 37 °C for 1 h. After incubation, solutions were transferred from the buckets into test tubes, followed by centrifugation at 2800 rpm for 10 min in order to precipitate red blood cells. The optical density of the obtained supernatants was measured using the GENESYS 6 spectrophotometer (Thermo, Waltham, MA, USA) at a wavelength of 545 nm. Positive and negative controls were double distilled water and saline, respectively. Haemolysis was measured as a sample-to-positive control ratio.

### 2.8. Platelet Aggregation Testing

To measure graft-induced platelet aggregation, the whole peripheral blood withdrawn from healthy volunteers was used. No aggregation inducers were used for platelet aggregation experiments. The blood was mixed with 3.8% sodium citrate at a ratio of 1:9 (citrate:blood). To obtain platelet-rich plasma (PRP), the citrated blood was further centrifuged at 1000 rpm for 10 min. Platelet-poor plasma (PPP) was obtained by re-centrifuging PRP at 4000 rpm for 20 min. PPP was used to calibrate the instrument, a semi-automatic 4-channel platelet aggregation analyser APACT 4004 (LABiTec). Intact pure PRP was used as a baseline for platelet aggregation. 25 µL 0.025M CaCl_2_ was added to 250 µL PRP to restore the level of Ca^2+^ in citrated blood. Grafts were exposed to CaCl_2_-supplemented PRP for 3 min followed by measurement of the maximum aggregation value on platelet analyser [15,16,17].

### 2.9. Thrombogenicity Testing

A method for measuring platelet adhesion to graft surface was based both on visual examination of platelet deformation and quantification of platelets [17]. Tubular grafts were converted to 2D by making an incision along their length. Graft samples (*n* = 3 samples per group, 0.5 cm^2^) were incubated in 300 μL PRP at 37 °C for 2 h. Samples were then washed with PBS to remove unadsorbed plasma components, fixed in a 2% glutaraldehyde solution for 2 h, then washed again with PBS, dehydrated in ascending concentrations of ethanol (30%, 50%, 70%, 80% and 95%, 15 min each) and finally dried at RT. Samples were sputter coated with gold/palladium (Au/Pd) using an Emitech SC 7640 vacuum post (Quorum Technologies, Lewes, UK).

Platelet adhesion was assessed using a S-3400N scanning electron microscope (Hitachi, Chiyoda, Japan) at 5 kV voltage under high vacuum at ×2000 magnification. Nine representative fields of view were randomly selected and platelet adhesion was calculated using the platelet deformation index [15,17]:

Deformation index = (number of type I platelets × 1 + number of type II platelets × 2 + number of type III platelets × 3 + number of type IV platelets × 4 + number of type V platelets × 5)/total platelet count. Types of platelet deformation are presented in Table 1.

### 2.10. Isolation and Characterization of Human ECFCs

To obtain human ECFCs, the whole peripheral blood was withdrawn from healthy volunteers and cell isolation was conducted according to the protocol established by our group previously [18]. Briefly, peripheral blood mononuclear cells were isolated using a Histopaque-1077 (Sigma) in accordance with the manufacturer’s instructions. Cells were washed twice with PBS, resuspended in EGM-2MV medium (Lonza) supplemented with 5% FBS (Hyclone), 2% penicillin/streptomycin (Invitrogen) and 0.25 g/mL amphotericin B (Invitrogen) and seeded in collagen-coated culture plates.

In the first 48 h, medium was changed daily to remove non-adherent cells and debris. After 7 days of culture, cells were transferred to fibronectin-coated plates and further cultured until 70% confluency. Visual monitoring of cultures was performed daily.

Immunophenotyping of ECFCs was performed by flow cytometry at the 3rd passage. Approximately 0.5–1 × 10^5^ cells were lifted, washed with PBS and stained by antibodies. Red blood cells were lysed by VersaLyse (Beckman Coulter, Brea, CA, USA). We used a combination of conjugated monoclonal antibodies from Biolegend (unless otherwise indicated): fluorescein isothiocyanate (FITC)–CD3 (Beckman Coulter, A07746), CD34 (Beckman Coulter, IM1870U), vWF (Abcam, ab8822); phycoerythrin (PE)—KDR (BD Biosciences, 560494), CD14 (Beckman Coulter, A07764); allococycyanin (APC)—CD133 (MACS, 130-090-826), CD31 (BioLegend, 303115); phycoerythrin-cyanine 7 (PC7)—CD146 (BioLegend, 361008); PacificBlue 450 (PB 450)—HLA-DR (BioLegend, 307633); Krome Orange (KrOr)—CD45 (Beckman Coulter, A96416).

We developed two fluorescence-activated cell sorting (FACS) panels for cell characterization:
CD3, CD14, HLA-DR, CD45;CD34, KDR, CD146, CD133, CD31, CD45;vWF, CD146.


Antibodies against proteins of interest or corresponding isotype controls (2 to 20 μL) were added to cells with the following incubation for 30 min at RT in dark. For intracellular vWF staining, cells were fixed and permeabilized using the IntraPrep kit (Beckman Coulter). Stained samples were analysed using CytoFlex instrument (Beckman Coulter) supplied with CytExpert software. To exclude cell doublets and debris, the target gate was restricted by FSC-A and FSC-H with transfer to the FSC/SSC histogram for the subsequent gating steps. The mean fluorescence intensity and percentage of cells positive for each marker were recorded.

### 2.11. Assessment of Viability and Adhesion of ECFCs Seeded onto the Graft Surface

Tubular grafts were converted to 2D by making an incision along their length. Sterile prostheses with and without RGD peptides (*n* = 5 samples per group) were fixed to the bottom of sterile 24-well culture plates using a 0.6% agarose solution. 2.5 × 10^5^ ECFCs of the 3rd passage were added into the plates and cultured for 72 h. The total number of cells per 1 mm^2^ surface and the relative proportion of dead cells on grafts were assessed using the fluorescence microscopy by staining cultured cells with 0.03 mg/mL ethidium bromide (Sigma) and 2 μg/mL Hoechst 33,342 (Sigma) 3 min before imaging. Eight fields of view per group were analysed.

Samples were then fixed in 1% glutaraldehyde (Sigma) for 1.5 h, followed by incubation with 1% osmium tetroxide (Serva Electrophoresis, Heidelberg, Germany) for 3 h. Samples were dehydrated for 15 min in ascending concentrations of ethanol (50%, 70%, 80% and 90%, 15 min each; 2× 95%, 30 min). Slides were dried at 37 °C and were sputter coated with gold/palladium (Au/Pd) using an Emitech SC 7640 vacuum post (Quorum Technologies). Graft surface was visually examined by backscattered electron microscopy (S-3400N, Hitachi, Chiyoda, Japan) at 15 kV voltage under high vacuum.

### 2.12. Assessment of Graft In Vivo Performance

All animal work was performed in accordance with European Convention for the Protection of Vertebrate Animals (Strasbourg, 1986). All surgical procedures were performed under sterile conditions. Rats were received from the Animal Core Facility of the Research Institute for Complex Issues of Cardiovascular Diseases. For graft implantation, 6 month old male Wistar rats weighing 400–450 g (*n* = 96) were anesthetized by inhalation of 3% isoflurane which was then maintained at a concentration of 1.5%. Vascular grafts with a diameter of 1.5 mm and a length of 10 mm were sterilized by ethylene oxide and implanted into rat abdominal aortas by end-to-end anastomosis. After surgery, all animals were returned to *ad libitum* access to food and water. Rats were sacrificed by an intraperitoneal injection of a sodium pentobarbital (100 mg/kg body weight) 1 and 3 months post operation (*n* = 6 animals per time point per group; the total number of rats with implanted RGD-modified grafts was 84, the total number of rats with implanted unmodified grafts was 12).

### 2.13. Histological Examination

Graft samples were fixed in 10% neutral phosphate buffered formalin (BioVitrum, Moscow, Russia) for 24 h, then washed in running tap water for 2 h, dehydrated in 5 changes of isopropanol (30 min per change). Sample embedding into paraffin (Histomix, BioVitrum) was performed at 56 °C in 3 changes (1 h per change). Samples were then glued on the block with the further sectioning (8 μm thickness) using a rotary microtome (Microm HM 325, Thermo Scientific, Waltham, MA, USA). Samples were dried overnight at 37 °C. Immediately before staining, samples were deparaffinized in xylene (3×, 5 min per change) with the following rehydration in 95% ethanol (3×, 5 min per change).

For haematoxylin and eosin staining, deparaffinized sections were rinsed with double distilled water and stained with a Harris haematoxylin (BioVitrum) for 5 min, followed by the dip in double distilled water and bluing in running tap water for 5 min. Subsequently, sections were immersed in differentiation solution (1% HCl dissolved in 70% ethanol) for 2 s, blued in running tap water for 30 min, rinsed with double distilled water and then stained with eosin (BioVitrum) for 30 s. After the rinse in double distilled water, sections were dehydrated in ascending concentrations of ethanol (70% for 10 s, 95% for 10 s and 95% for 1 min), cleared with xylene for 3 min and mounted using VitroGel (BioVitrum).

For Van Gieson staining, deparaffinized sections were immersed in 95% ethanol (3×, 5 min), rinsed with double distilled water and stained with Weigert’s haematoxylin (BioVitrum) for 10 min. After the bluing in running tap water for 3 min, sections were immersed in differentiation solution (1% HCl dissolved in 70% ethanol) for 2 s, blued in running tap water for 30 min, rinsed with double distilled water and then stained with picrofuchsin (BioVitrum) for 2.5 min followed by rinse in double distilled water, dehydration in ascending concentrations of ethanol (70% for 10 s, 95% for 10 s and 95% for 1 min), clearance with xylene for 3 min and mounted using VitroGel (BioVitrum).

All slides were examined using the AxioImager.A1 light microscope (Carl Zeiss, Oberkochen, Germany).

### 2.14. Evaluation of Graft Calcification

To assess the presence of calcium, graft sections were stained with alizarin red S (KhimService, Novomoskovsk, Russia) and DAPI (Sigma). Sections were immersed in 2% aqueous solution of alizarin red S for 70 seconds with the further counterstaining of nuclei with DAPI (10 μg/mL) for 3 min. The specimens were then washed with double distilled water, dried and mounted. Samples were examined using the AxioImager.A1 light microscope (Carl Zeiss).

### 2.15. Immunofluorescence Staining

Sections (8 μm thickness) were cut using the cryostat (Microm HM 525, Thermo Scientific). Immunofluorescence staining was performed using the following primary antibodies: mouse anti-CD31 (ab119339, Abcam, Cambridge, UK), rabbit anti-CD34 (ab185732, Abcam), mouse anti-collagen type I (ab6308, Abcam), rabbit anti-collagen type IV (ab6586, Abcam) or FITC-conjugated von Willebrand factor (ab8822, Abcam). The following secondary antibodies were used: goat anti-mouse Alexa Fluor 568 conjugate (ab175473, Abcam) and donkey anti-rabbit Alexa Fluor 488 conjugate (ab150073, Abcam). Autofluorescence was minimized with the Autofluorescence Eliminator Reagent (Millipore) according to the manufacturer’s instructions. All sections were counterstained with DAPI (Sigma) and mounted with ProLong Gold Antifade (Life Technologies, Carlsbad, CA, USA). Staining was visualized by confocal microscopy (LSM 700, Carl Zeiss, Oberkochen, Germany).

### 2.16. Statistical Analysis

Statistical analysis was performed using GraphPad Prism 7 (GraphPad Software, San Diego, CA, USA). Normality of distribution was assessed by Kolmogorov-Smirnov test. Two independent groups were compared by Mann-Whitney U-test while three or more independent groups were compared using Kruskal-Wallis test with Dunn’s post hoc test. *p* values of ≤0.05 were regarded as statistically significant. Data are presented as the median and interquartile range.

## 3. Results

### 3.1. Fabrication of Small-Diameter Vascular Grafts in Different Modes of RGD Peptide Modification

Peptides immobilize on graft surface through linker groups such as amines. To this end, our first step of graft modification was partial aminolysis of ester bonds of the graft material using two hydrophilic linkers of different length, short (1,6-hexamethylenediamine, Amine1) and long (4,7,10-trioxa-1,13-tridecanediamine, Amine2), hypothesizing that amine length may influence biocompatibility (Figure 1A,B).

Modified grafts displayed amino group density ranging from 5.8 × 10^−9^ to 8.9 × 10^−9^ mol/cm^2^ (Table 2), in agreement with the numbers reported by others. Optimization of reaction time showed that the number of amino groups peaked after 60 min for Amine1 and 30 minutes for Amine2 (Table 2) and these time points were further used for the peptide immobilization.

During the second step of modification, RGD-containing peptides were immobilized on graft surface by treatment with glutaraldehyde and sodium cyanoborohydride. Complete sequences of RGD peptides are presented in Figure 1A. Pep1 and Pep3 were attached to the graft through the ε-amino group of lysine, whist Pep2 was bound to the amino group of 6-aminohexanoic acid. Peptide immobilization was confirmed by Sakaguchi test for arginine. Eventually, 6 different modifications of biodegradable vascular grafts were fabricated (Table 3).

### 3.2. Tensile Testing of RGD-Modified Grafts

We then sought to examine the physical properties of resultant grafts. In comparison with unmodified prostheses, we observed a substantial decrease in the tensile strength of RGD-modified grafts regardless of the amine or peptide used (Table 4). As compared to unmodified prostheses, ultimate tensile strength of Amine1- and Amine2-treated grafts decreased on average 4.1- and 3.2-fold, respectively (Table 4). Elastic properties measured as elongation at break and Young’s modulus did not significantly differ between Amine2-treated and unmodified grafts (Table 4). However, Amine1-treated prostheses exhibited a substantial drop in Young’s modulus for Pep2 and Pep3 samples (Table 4).

Thus, different modes of RGD modification can affect tensile properties of polymer grafts. Notably, the use of Amine2 with shorter aminolysis time (30 min) enabled retaining the mechanical parameters.

### 3.3. Modification by All Types of RGD Peptides Negatively Affects Graft Hemocompatibility

We then questioned whether modification by RGD peptides induces haemolysis or platelet aggregation in vitro. Upon contact of the whole peripheral blood with grafts, the proportion of lysed red blood cells did not exceed 1% in all studied groups indicative of negligible haemolysis (Table 5). Exposure of RGD-modified and unmodified prostheses to the human PRP revealed that almost all RGD-modified grafts provoked higher platelet aggregation while unmodified grafts exhibited thrombogenicity similar to the pure PRP (Table 5). We then employed scanning electron microscopy to examine the platelet deformation by calculating a semi-quantitative index (Figure 2). In comparison with unmodified prostheses, we observed a significantly higher platelet deformation index for Amine1Pep2 and Amine1Pep3 grafts (Table 6). The surface of these grafts displayed multiple large clots of blood proteins and greater number of platelets with a predominance of type III-IV platelets (Table 6).

Taken together, these data suggest that regardless of the peptide or linker arm used, unmodified grafts are on average less thrombogenic than those modified by RGD-containing peptides.

### 3.4. Graft Modification with Amine2Pep3 Improves Adhesion and Viability of Human ECFCs In Vitro

ECFCs are a subset of endothelial progenitor cells exhibiting a considerable angiogenic and proliferative potential [19,20], therefore being involved in neovascularization and re-endothelialisation of damaged vessels [21,22]. To investigate endothelialisation of studied grafts in vitro, we differentiated human ECFCs from peripheral blood mononuclear cells and extensively characterized them by flow cytometry. Based on FACS profiling, the isolated cells indeed corresponded to ECFC immunophenotype (CD31^+^vWF^+^KDR^+^CD146^+^) and were concurrently negative for hematopoietic immune cell markers such as CD3, CD14, CD45 and HLA-DR (Table 7).

Grafts modified by Amine2Pep3 demonstrated the highest number of adhered ECFCs (2682 cells/mm^2^) whilst other prostheses had 6 times fewer adherent cells (Figure 3A,B). The total number of dead cells on the Amine2Pep3 graft was also the highest as compared to other RGD-treated prostheses (222 cells/mm^2^, Figure 3A,B). The percentage of viable cells on all RGD-treated grafts exceeded 90%, whereas only around 20% cells were viable in unmodified prostheses (Figure 3C). Accordingly, the proportion of dead cells in unmodified grafts was 76.5% but did not exceed 9% in RGD-treated prostheses (Figure 3D). Scanning electron microscopy analysis confirmed adhesion of ECFCs forming a monolayer on the luminal surface of RGD-treated but not unmodified grafts, particularly Amine2Pep3 (Figure 4).

Collectively, these data indicate that the use of Amine2Pep3 combination for graft modification significantly improves ECFC adhesion in vitro.

### 3.5. Modification with Amine2Pep3 Enhances Graft Biocompatibility In Vivo

We then implanted 1.5-mm-diameter RGD-treated and unmodified prostheses into rat abdominal aortas to compare their in vivo performance. 1 and 3 months after implantation, we evaluated primary patency rate, endothelialisation, extracellular matrix deposition and calcification.

Both 1 and 3 months post operation, 2 of 6 (33.3%) unmodified grafts were occluded. Remaining unmodified implants were characterized by moderate cellularity with most cells residing in the outer part of the prosthesis wall (Figure 5A,B). 3 months post implantation, half of patent grafts contained calcium deposits within the neointima (Figure 5A,B, Table 8). In RGD-modified grafts, cells migrated from the outer to the luminal part of the implant wall (Figure 5A,B). We also detected abundant extracellular matrix throughout the graft wall, with more pronounced collagen deposition near the luminal surface, which also demonstrated signs of neointimal hyperplasia (Figure 5A,B).

With regard to the distinct RGD peptides, Amine2Pep1 grafts demonstrated a better primary patency (6/6, 100%) compared to Amine1Pep1 implants (4/6, 66.7%), which frequently showed thrombosis at the anastomosis site (Figure 5A,B, Table 8). Likewise, Amine2Pep1 grafts were characterized by substantially higher cellularity after 1 and 3 months of implantation (Figure 5A,B). Calcification was also retarded in Amine2Pep1 grafts, being observed 3 months post implantation (2/6, 33.3%), whereas Amine1Pep1-modified grafts showed similar calcification rate as early as 1 month after the implantation (Figure 5A,B, Table 8).

All Pep2-modified grafts exhibited complete primary patency regardless of the linker group used but evinced poor cellularity excepting 1-month time point in Amine1Pep2 group (Figure 5A,B, Table 8). Calcification loci associated with neointima were detected in half of these grafts both 1 and 3 months post implantation (Figure 5A,B, Table 8).

Similarly, all Pep3-treated grafts displayed 100% patency irrespective of the amine linker employed (Figure 5A,B, Table 8). Amine1Pep3 prostheses were characterized by moderate number of cells at both time points (Figure 5A,B). Implants with immobilized Amine2Pep3 were well-infiltrated by cells 1 month post operation, with the further increase in cellularity at 3 months (Figure 5A,B). With regards to calcification, grafts modified with Amine1Pep3 were more frequently calcified (50%) in comparison with Amine2Pep3 (33.3%) at 3-month time point (Figure 5A,B, Table 8).

We then compared endothelialisation of RGD-modified grafts by means of CD31^+^/CD34^+^ cell ratio determined by immunofluorescence examination. The resulting score reflected to a certain extent the proportion of mature endothelial cells in relation to endothelial progenitor cells and therefore served as a semi-quantitative measure of endothelialisation within implants. We found that grafts modified by Amine2 linker arm exhibited a substantial endothelialisation at both time points and this effect was particularly evident for Amine2Pep1 and Amine2Pep3 grafts 3 months post implantation (Figure 6A–D). The Amine1 linker arm did not promote endothelialisation in comparison with unmodified prostheses (Figure 6A–D). In addition, different modes of RGD modification did not affect the expression of basement membrane components type IV collagen and vWF (Appendix A).

To summarize, these findings identify Amine2Pep3 modification as an optimal formula for modification of the grafts in terms of primary patency, adequate cellularity and absence of early calcification.

## 4. Discussion

Arterial replacement is frequently used for treatment of vascular disease including atherosclerosis. Autologous blood vessels such as saphenous vein or internal thoracic artery are a gold standard for bypass surgery; however, their availability is often limited due to pre-existing vascular disease or vein stripping [23]. In addition, harvesting these vessels is invasive and can be associated with risk of complications [23]. Small-diameter vascular grafts fabricated from synthetic polymer materials are utilized in clinical practice, yet their poor biocompatibility resulting in thrombosis that demands repeated surgical interventions remains a major problem [3]. Thus, development of biocompatible small-calibre vascular grafts is a recognized but unmet clinical need.

Here, we carried out a biocompatibility testing of PHBV/PCL-based polymer grafts modified by different RGD sequence peptides immobilized using short and long amine linkers.

All modes of RGD modification reduced ultimate tensile strength of the grafts, probably due to aggressive chemicals necessary for immobilization; however, Amine2 (4,7,10-trioxa-1,13-tridecanediamine) required shorter aminolysis time, thus enabling to retain elasticity of the bioartificial conduit. Neither different amine linker arms nor RGD peptides affected haemolysis upon the contact with modified grafts in vitro, however, all the RGD-treated grafts displayed a tendency to promote platelet aggregation in comparison with unmodified counterparts. This result could indicate that RGD peptides or amine linkers promote thrombogenicity in vitro. The best results for cell adhesion were demonstrated for Amine2Pep3 prostheses. This difference can be explained by an insufficient length of 1,6-hexamethylenediamine, whereas longer linker 4,7,10-trioxa-1,13-tridecanediamine could enable better presentation of RGD peptides for the interaction with cell receptors.

Other groups studied the performance of polymer vascular grafts modified by RGD peptides co-immobilized with heparin [24], VEGF [25] and K5 peptide [26], without inquiring if RGD sequence per se may affect implant biocompatibility. Some rather limited evidence suggested that cyclic RGD peptides may have higher affinity to integrins than linear ones, thus contributing to better cell attachment [27]. In this study we for the first time provided a side-by-side comparison of cyclic and linear RGD peptides immobilized by linker arms of a different length. We demonstrated that Amine2Pep3 grafts substantially promoted ECFC adhesion as compared to other RGD-linker combinations or unmodified prostheses. In addition to the effect of Pep3 itself, this result was partially influenced by Amine2, because the Amine1Pep3 counterpart exhibited modest biocompatibility results. However, Amine2Pep3 did not significantly improve graft hemocompatibility as assessed by platelet aggregation assay, displaying higher platelet adhesion in comparison with unmodified grafts. In agreement with in vitro results, short-term implantation of RGD-modified vascular grafts into abdominal aortas of rats revealed a more pronounced regeneration of vascular tissues and better overall biocompatibility of Amine2Pep3 prostheses in comparison with other modes of RGD modification or unmodified implants.

Calcification is one of the major causes of prosthetic blood vessel dysfunction [28,29]. Formation of calcium deposits not only occurs in autologous vessel conduits but also affects synthetic grafts. For example, ePTFE prostheses were shown to undergo calcification upon the implantation [30]. RGD peptides might be used for tissue-engineered bone graft fabrication, as mesenchymal stem cells switched to osteoblastic differentiation upon culture in RGD-modified hydrogels [31]. In the present study, only 2 of 12 Amine2Pep3-implanted rats showed evidence of calcification in a short-term period, suggesting this combination as an optimal graft modification formula. The highest rate of calcification was observed for Pep2 (10/24 rats with intimal calcification). These opposite examples highlight the importance of RGD motif in altering the graft capability to calcify.

The major challenge in developing synthetic small-diameter vascular grafts is to mimic the properties of the native vessel. An ideal material for such grafts should be remodellable by host cells and its degradation products must be easily metabolized and excreted by the organism. Modification of prostheses with bioactive molecules needs to address multiple issues, including control of platelet adhesion, coagulation cascade and thrombus formation, whilst promoting endothelialisation. Peptides with different RGD sequence as well as their linkers are completely unstudied in terms of vascular graft modification. The results presented in our study provide an insight on the functionality of small-diameter vascular prostheses in different modes of RGD modification. Further in-depth trials are required to validate our results in patients requiring arterial replacement.

## 5. Conclusions

In summary, here we compared the performance of small-calibre biodegradable vascular grafts in different modes of RGD modification. We conclude that 1.5-mm-diameter PHBV/PCL prosthesis modified with cyclic Arg-Gly-Asp-Phe-Lys immobilized by a long linker 4,7,10-trioxa-1,13-tridecanediamine (Amine2Pep3) exhibits superior biocompatibility and is fully suitable for clinical testing.

## Figures and Tables

**Figure 1 polymers-11-00174-f001:**
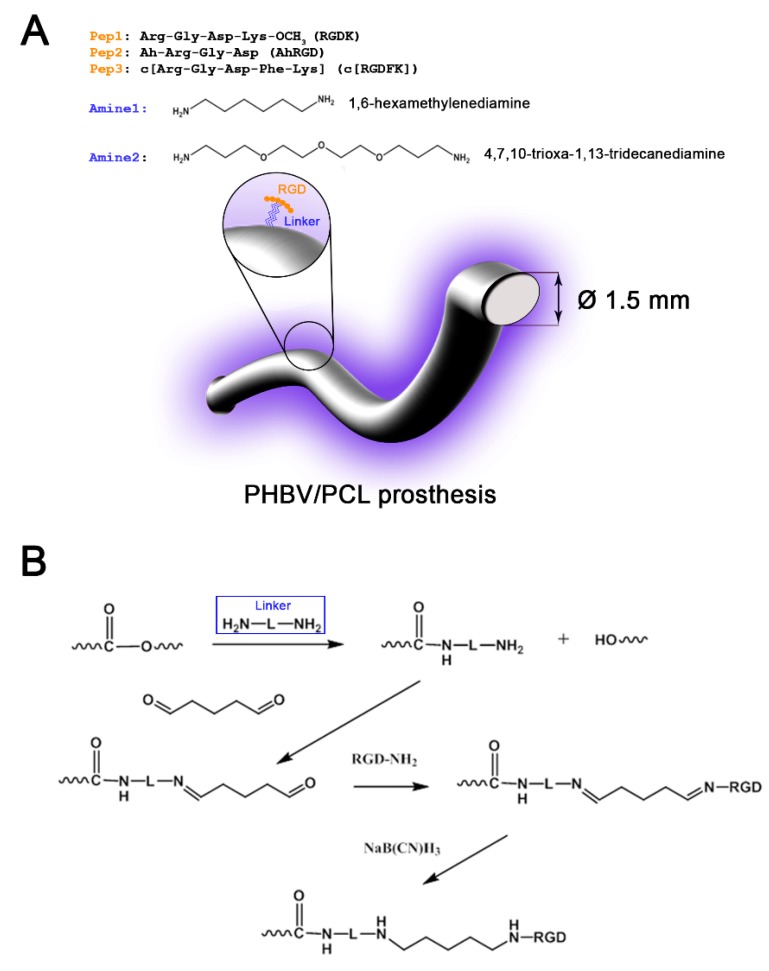
Study design. (**A**) A cartoon illustrating the modification of poly(3-hydroxybutyrate-*co*-3-hydroxyvalerate)/poly(ε-caprolactone) vascular grafts with RGD-containing peptides. Listed are RGD peptides used to modify the luminal surface of grafts and linker arm groups utilized for peptide immobilization. (**B**) General scheme for the graft surface modification, where H_2_N–L–NH_2_ is a linker group.

**Figure 2 polymers-11-00174-f002:**
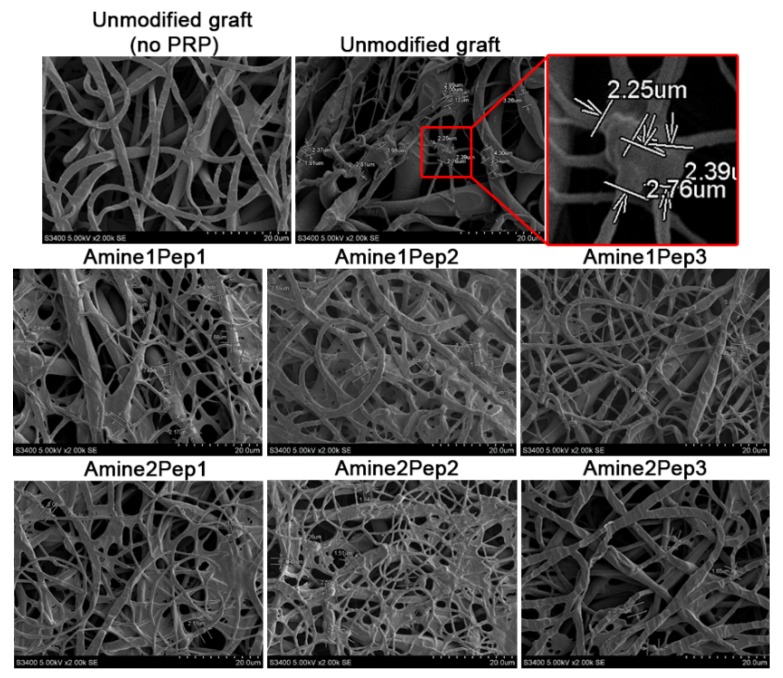
Representative scanning electron microscopy images of grafts functionalized with distinct RGD peptides or unmodified prostheses upon the contact with human platelet-rich plasma (PRP).

**Figure 3 polymers-11-00174-f003:**
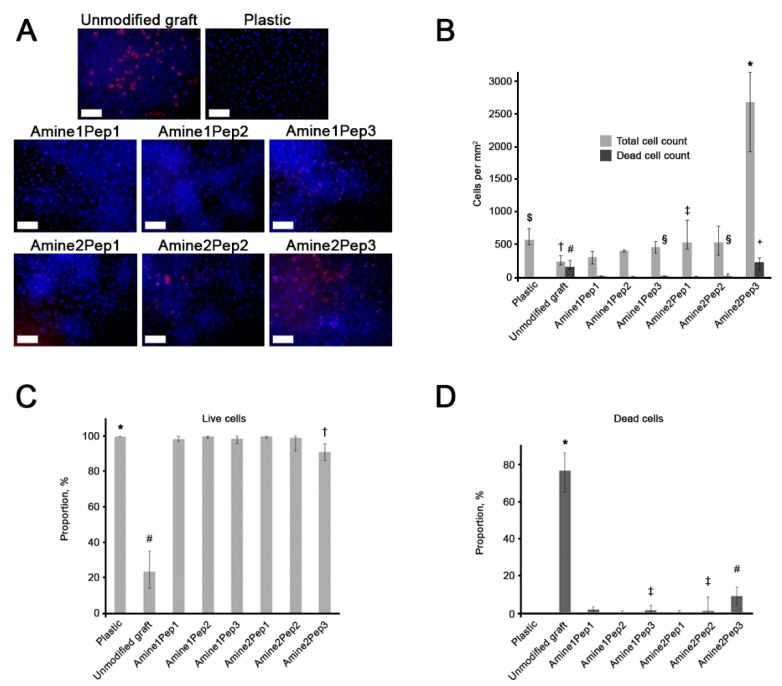
Testing of adhesion and viability of endothelial cells on grafts functionalized with distinct RGD peptides or unmodified prostheses. (**A**) Representative images of endothelial colony-forming cells (ECFCs) adherent to the surface of grafts and stained with ethidium bromide (red, stains dead cells) and Hoechst 33,342 (blue, stains all cells). Scale bar = 100 μm. (**B**) Quantification of cells from experiment in A. Whiskers indicate standard deviation. $ *p* < 0.05 in comparison with unmodified grafts, Amine1Pep1, Amine1Pep2, Amine2Pep3; † *p* < 0.05 in comparison with a culture plastic, Amine1Pep2, Amine1Pep3, Amine2Pep1, Amine2Pep2, Amine2Pep3; ‡ *p* < 0.05 in comparison with all other study groups; # *p* < 0.05 in comparison with all other study groups excepting Amine2Pep3; + *p* < 0.05 in comparison with all other study groups excepting unmodified grafts; § *p* < 0.05 in comparison with a culture plastic. (**C**) Relative proportion of live ECFCs on the surface of studied grafts after 72 h of culture. Whiskers indicate standard deviation. * *p* < 0.05 in comparison with unmodified grafts, Amine1Pep1, Amine1Pep3, Amine2Pep2, Amine2Pep3; # *p* < 0.05 in comparison with all other study groups; † *p* < 0.05 in comparison with a culture plastic, unmodified grafts, Amine1Pep2, Amine2Pep1. (**D**) Relative proportion of dead ECFCs on the surface of studied grafts after 72 h of culture. Whiskers indicate standard deviation. * *p* < 0.05 in comparison with all other study groups; ‡ *p* < 0.05 in comparison with a culture plastic; # *p* < 0.05 in comparison with a culture plastic, Amine1Pep2, Amine2Pep1.

**Figure 4 polymers-11-00174-f004:**
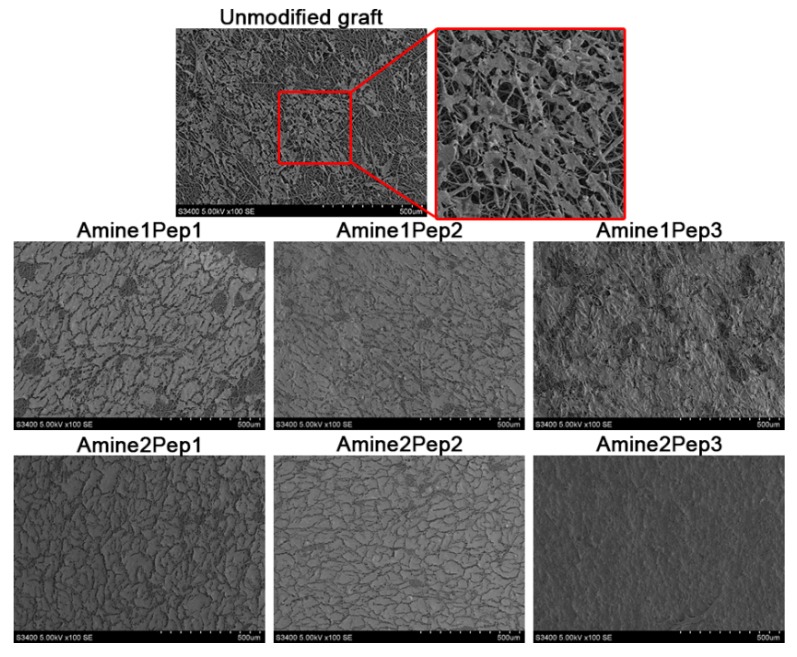
Representative scanning electron microscopy images of endothelial colony-forming cells (ECFCs) adhered to the surface of grafts functionalized with distinct RGD peptides or unmodified prostheses after 72 h of culture.

**Figure 5 polymers-11-00174-f005:**
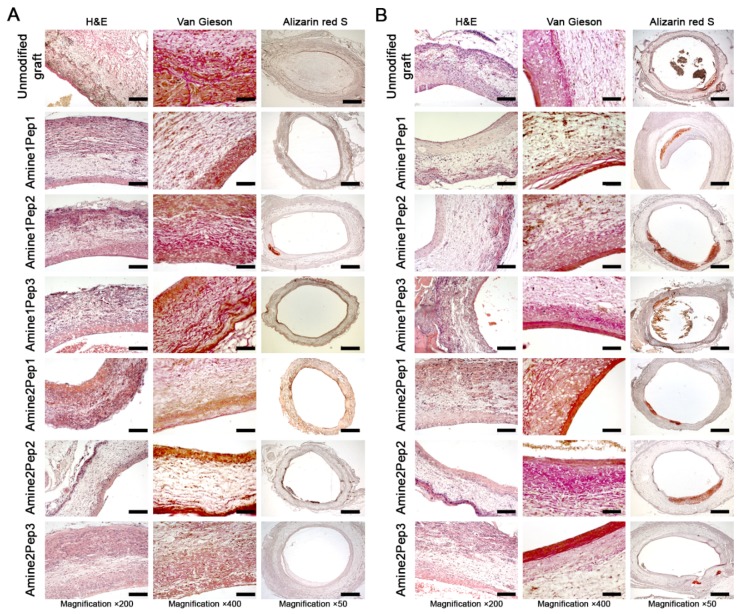
Histological examination of grafts functionalized with distinct RGD peptides or unmodified prostheses implanted into rat abdominal aortas for 1 (**A**) and 3 (**B**) months. Haematoxylin and eosin (scale bar = 100 μm), van Gieson (scale bar = 50 μm) and alizarin red S (scale bar = 500 μm) staining.

**Figure 6 polymers-11-00174-f006:**
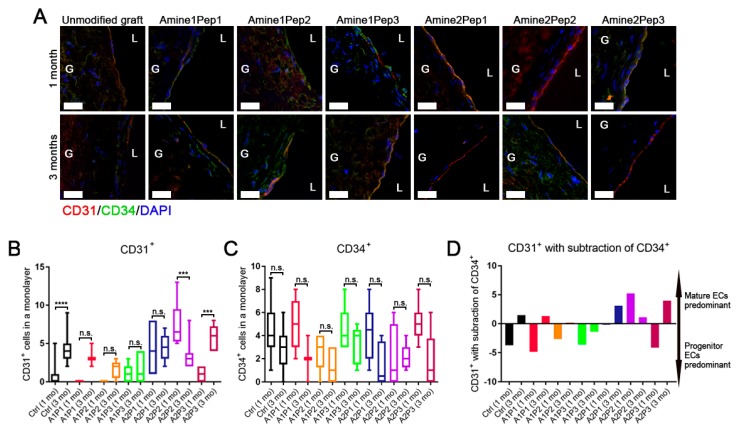
Immunofluorescence assessment of endothelialisation of grafts functionalized with distinct RGD peptides or unmodified prostheses. (**A**) Double immunostaining for CD31 (red, mature endothelial cells) and CD34 (green, endothelial progenitor cells) with DAPI (blue, nuclei) counterstaining, representative confocal microscopy images. Scale bar = 20 μm. G means graft, L means lumen. (**B**) CD31^+^ cell count. Whiskers indicate range, boxes bounds indicate 25th and 75th percentiles, centre lines indicate median. One-way ANOVA with Tukey’s multiple comparisons test. (**C**) CD34^+^ cell count. Whiskers indicate range, boxes bounds indicate 25th and 75th percentiles, centre lines indicate median. One-way ANOVA with Tukey’s multiple comparisons test. (**D**) Semi-quantitative analysis of endothelial phenotype based on CD31^+^ and CD34^+^ cell count.

**Table 1 polymers-11-00174-t001:** Types of platelet deformation.

Type	Platelet Features
I	Disc-shaped (no deformation)
II	Platelet is increased in size with protrusion-like pseudopodia sticking out
III	Platelet is substantially increased in size, irregularly shaped, with pronounced pseudopodia, multiple platelets aggregate together
IV	Spreading of the platelet, cytoplasm expands among pseudopodia
V	Platelet in the form of a spot with granules, pseudopodia cannot be identified due to cytoplasm spreading

**Table 2 polymers-11-00174-t002:** The number of amino groups on poly(3-hydroxybutyrate-*co*-3-hydroxyvalerate)/poly(ε-caprolactone) vascular grafts after aminolysis.

Time, min	The Number of Amino Groups *, mol/cm^2^	
	Amine1 (1,6-hexamethylenediamine)	Amine2 (4,7,10-trioxa-1,13-tridecanediamine)
**10**	5.1 × 10^−9^	5.8 × 10^−9^
**30**	6.3 × 10^−9^	8.6 × 10^−9^
**60**	8.9 × 10^−9^	8.1 × 10^−9^

* Based on results from three independent experiments.

**Table 3 polymers-11-00174-t003:** Fabrication of vascular grafts in different modes of RGD modification.

Linker	Peptide 1 (RGDK)	Peptide 2 (AhRGD)	Peptide 3 (c[RGDFK])
**Amine1 (1,6-hexamethylenediamine)**	Amine1Pep1	Amine1Pep2	Amine1Pep3
**Amine2 (4,7,10-trioxa-1,13-tridecanediamine)**	Amine2Pep1	Amine2Pep2	Amine2Pep3

**Table 4 polymers-11-00174-t004:** Tensile testing of vascular grafts functionalized with distinct RGD peptides or unmodified prostheses.

Sample	*n*	Ultimate Tensile Strength, MPa Median (25th and 75th Quartiles) Range	Elongation at Break, % Median (25th and 75th Quartiles) Range	Young’s modulus, MPa Median (25th and 75th Quartiles) Range	Thickness, mm Median (25th and 75th Quartiles) Range
**Internal Mammary Artery**	6	2.48 (1.36–3.25) 1.07–6.52	29.72 (23.51–39.62) 22.0–50.88	2.42 (1.87–3.19) 1.53–3.34	0.27 (0.24–0.30) 0.21–0.90
**Unmodified Grafts**	6	3.85 (2.88–4.54) 2.38–4.62	102.7 (79.37–106.3) 74.92–119.23	21.8 (19.2–25.2) 18.2–27.5	0.38 (0.35–0.46) 0.35–0.47
**Amine1Pep1**	6	**1.29 (0.65–1.42)** **0.51–1.58 ***	127.39 (31.5–135.52) 17.23–182.52	21.4 (10.2–24.8) 8.46–25.3	**0.55 (0.49–0.55)** **0.49–0.56 ***
**Amine1Pep2**	6	**0.81 (0.77–1.1)** **0.68–1.2 ***	83.76 (81.2–132.8) 31.47–154.3	**11.8 (11.1–17.5)** **9.64–18.3 ***	0.45 (0.45–0.5) 0.41–0.56
**Amine1Pep3**	6	**0.72 (0.45–0.89)** **0.3–1.0 ***	84.0 (17.26–99.83) 9.19–148.9	**8.11 (6.86–12.4)** **5.05–12.7 ***	0.45 (0.44–0.47) 0.42–0.5
**Amine2Pep1**	6	**1.2 (1.14–1.22)** **1.1–1.56 ***	65.73 (65.03–121.46) 27.53–155.63	22.1 (21.5–24.6) 20.7–24.9	**0.54 (0.53–0.54)** **0.53–0.58 ***
**Amine2Pep2**	6	**1.21 (1.21–1.23)** **1.14–1.26 ***	109.93 (93.28–127.23) 86.47–132.54	22.9 (21.1–23.1) 21.0–23.3	**0.49 (0.49–0.53)** **0.49–0.55 ***
**Amine2Pep3**	6	**1.18 (1.03–1.6)** **1.02–1.79 ***	136.74 (84.43–181.15) 76.32–198.09	19.4 (18.6–24.5) 17.9–27.6	**0.52 (0.49–0.53)** **0.49–0.53 ***

* *p* < 0.05 compared to unmodified grafts.

**Table 5 polymers-11-00174-t005:** In vitro evaluation of haemolysis and platelet aggregation upon the contact with vascular grafts functionalized with distinct RGD peptides or unmodified prostheses.

Sample	Haemolysis, % Median (25th and 75th Quartiles) Range	Maximum Platelet Aggregation, % Median (25th and 75th Quartiles) Range
**Pure Platelet-rich Plasma** **(Baseline Platelet Aggregation)**	-	14.61 (13.63–17.72) 9.43–20.64
**Unmodified Grafts**	0 (0–0) 0–0	17.25 (16.3–17.96) 15.89–18.63
**Amine1Pep1**	0.36 (0–0.72) 0–0.72	**24.09 (24.09–25.65)** **23.44–26.28 ***
**Amine1Pep2**	0.36 (0.36–0.72) 0.36–0.72	20.12 (19.56–21.19) 15.34–23.44
**Amine1Pep3**	0.36 (0–0.36) 0–0.36	**22.54 (22.49–23.74)** **18.04–31.32 ***
**Amine2Pep1**	0.36 (0–0.72) 0–0.72	**21.58 (21.44–24.35)** **21.19–24.39 ***
**Amine2Pep2**	0.72 (0–1.08) 0–1.08	**20.74 (19.95–23.59)** **19.24–23.94 ***
**Amine2Pep3**	0.72 (0–0.72) 0–0.72	**22.24 (22.10–24.27)** **22.04–25.54 ***

* *p* < 0.05 compared to unmodified grafts.

**Table 6 polymers-11-00174-t006:** Measurement of the platelet deformation index and platelet type ratios based on scanning electron microscopy analysis of vascular grafts functionalized with distinct RGD peptides or unmodified prostheses.

Sample	Platelet Type Ratio, %					Platelet Deformation Index Median (25th and 75th Quartiles)
	I	II	III	IV	V	
**Unmodified Grafts**	0	15.4	73.1	9.6	1.9	2.7 (1.0–3.0)
**Amine1Pep1**	5.6	36.1	25	30.6	2.8	2.5 (2.0–3.0)
**Amine1Pep2**	**1.5**	**11.4**	**50**	**31.4**	**5.7**	**3.31 (3.0–3.7) ***
**Amine1Pep3**	**0**	**10.8**	**43.1**	**30.7**	**15.4**	**3.7 (3.4–4.5) ***
**Amine2Pep1**	3.1	31.3	28.1	15.6	21.9	2.6 (1.0–3.7)
**Amine2Pep2**	0	65.5	24.2	3.4	6.9	1.3 (0.0–2.2)
**Amine2Pep3**	4.8	28.6	40.5	7.1	19.0	2.9 (2.5–4.0)

* *p* < 0.05 compared to unmodified grafts.

**Table 7 polymers-11-00174-t007:** Phenotype profiling of the 3rd passage human endothelial colony-forming cells (ECFCs) by flow cytometry.

Markers	CD31, % Range	vWF, % Range	KDR, % Range	CD146, % Range	CD34, % Range	CD133, % Range	CD3, % Range	CD14, % Range	CD45, % Range	HLA-DR, % Range
3rd passage human ECFCs	99.8 (99.1–100)	94.1 (91.6–96.2)	60.2 (48.4–68.0)	99.5 (98.7–99.8)	2.2 (0.1–8.9)	0 (0–0.9)	0.0 (0.0–0.0)	0.0 (0.0–0.0)	0.0 (0.0–0.0)	0.0 (0.0–0.0)

**Table 8 polymers-11-00174-t008:** Primary patency, frequency of thrombosis and calcium deposition in vascular grafts functionalized with distinct RGD peptides or unmodified prostheses and implanted into rat abdominal aortas for 1 or 3 months.

Sample	Primary Patency		Thrombosis		Calcium Deposition	
	1 Month	3 Months	1 Month	3 Months	1 Month	3 Months
**Unmodified Grafts**	4/6	4/6	2/6	2/6	0/6	3/6
**Amine1Pep1**	4/6	4/6	2/6	2/6	2/6	2/6
**Amine1Pep2**	6/6	6/6	0/6	0/6	1/6	3/6
**Amine1Pep3**	6/6	6/6	0/6	0/6	0/6	3/6
**Amine2Pep1**	6/6	6/6	0/6	0/6	0/6	2/6
**Amine2Pep2**	6/6	6/6	0/6	0/6	3/6	3/6
**Amine2Pep3**	6/6	6/6	0/6	0/6	0/6	2/6

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
