# Peer review of "Biocompatibility of Small-Diameter Vascular Grafts in Different Modes of RGD Modification"

_polymers, 2019, doi:10.3390/polym11010174_

Round 1

Reviewer 1 Report

The authors investgated different  RGB/linker combinations for the functionalisation of a PHBV/PCL tubular electrospun scaffold. Intesrestingly, the authors performed a complete set of tests to verify and select the optimal biocompatible functionalised scaffolds. The investigations included mechanical properties, in vitro hemolysis, thrombogenicity, cytotoxicity and cell adhesion using human blood, platelets and endothelial cells as well as in vivo rat model for the implantation of the vascular grats to study endothelisation and calcification of the grafts up to 3 months. 

The authors present a well designed and soundly described study.

Minor comments:

the authors should specify if the mechanical properties were performed on the scaffolds that were sterilized. It is important to investigate if the sterilization process would modify the mechanical properties.

it is not clear why the statistical analysis of figure 6 is not presented. The authors should provide the statistical significance and comparison of the different peptide/linker combinations.

Author Response

Minor comments:

the authors should specify if the mechanical properties were performed on the scaffolds that were sterilized. It is important to investigate if the sterilization process would modify the mechanical properties.

Prior to mechanical tests, the samples were not sterilized. We included this information in Materials and Methods.

it is not clear why the statistical analysis of figure 6 is not presented. The authors should provide the statistical significance and comparison of the different peptide/linker combinations.

We have now provided statistical analysis for Figure 6.

We thank the reviewer for their useful comments.

Reviewer 2 Report

a.       The abstract seems short of information – The modified prostheses is tested for mechanical properties, hemocompatibility, thrombogenicity, and endothelialization. However, the only emphasis on biocompatibility was mentioned as the outcome of the results. The abstract concludes the markedly enhanced performance of the graft performance without proper comments on mechanical or other test results? Please add few more lines about these results in the abstract.

b.      Not being a biologist, a common question – why are 4-mm diameter prostheses used for in vitro and 1.5 mm used for in vivo? Just a short sentence justifying the selection would help the readers

c.       The tensile strengths decreased for modified grafts? Though no difference was found between amine 1 and 2 modifications, the tensile strength is lower than the internal mammary artery also, will this be justified for being really useful/applicable?

d.      Does it have to do with the thickness because there is a huge difference in thickness between unmodified and modified grafts and also with the mammary artery?

e.       Why is the thickness of modified grafts varying? Can it be controlled?

f.       The thrombogenic studies also conclude that the unmodified are less thrombogenic vs modified.  Please provide a hypothesis for your observations

g.      Again the authors fail to conclude if the results are right direction or not? Again fail to explain the results convincingly.

Author Response

a.       The abstract seems short of information – The modified prostheses is tested for mechanical properties, hemocompatibility, thrombogenicity, and endothelialization. However, the only emphasis on biocompatibility was mentioned as the outcome of the results. The abstract concludes the markedly enhanced performance of the graft performance without proper comments on mechanical or other test results? Please add few more lines about these results in the abstract.

We have substantially rewritten the abstract to address your comments.

b.      Not being a biologist, a common question – why are 4-mm diameter prostheses used for in vitro and 1.5 mm used for in vivo? Just a short sentence justifying the selection would help the readers

1.5-mm-diameter grafts represent an in vivo model for studying small-caliper prostheses.  4-mm-diameter grafts were used in vitro because they are much easier to handle (especially for conversion of tubular grafts into 2D) as well as because they are having an increased area enabling studies of cell adhesion and proliferation. We have now clearly explained this in paragraph 2.2. Graft Fabrication.

c.       The tensile strengths decreased for modified grafts? Though no difference was found between amine 1 and 2 modifications, the tensile strength is lower than the internal mammary artery also, will this be justified for being really useful/applicable?

All synthetic vascular grafts (fabricated from any polymer) are characterized by much lower tensile strength as compared to intact vessels, which contain multiple extracellular matrix molecules contributing to their elasticity. This is a well-known fact for specialists in the field, and it does not bring any usefulness per se.

d.      Does it have to do with the thickness because there is a huge difference in thickness between unmodified and modified grafts and also with the mammary artery?

Modified grafts in our study were expectedly thicker due to multiple chemical modifications. We do not believe sample thickness directly affects tensile strength in our study.

e.       Why is the thickness of modified grafts varying? Can it be controlled?

Thickness variation here comes from the electrospinning instrument which operates within the limits of permissible error. In other words, it is impossible to fabricate two different grafts with the same thickness due to the nature of the electrospinning technology.

f.       The thrombogenic studies also conclude that the unmodified are less thrombogenic vs modified.  Please provide a hypothesis for your observations

This is an observational comparative study assessing different types of RGD peptides and amine linkers with regards to their effect on graft performance. There was no hypothesis in this study as there was no proof-of-principle. With regard to your comment, we have added the following sentence to the discussion section:

“Neither different amine linker arms nor RGD peptides affected hemolysis upon the contact with modified grafts in vitro, however, all the RGD-treated grafts displayed a tendency to promote platelet aggregation in comparison with unmodified counterparts. This result could indicate that RGD peptides or amine linkers promote thrombogenicity in vitro.”

In vivo studies revealed no thrombi in rats with RGD-modified grafts.

g.      Again the authors fail to conclude if the results are right direction or not? Again fail to explain the results convincingly.

We are unsure how to address this point. The present study has no direction (right or wrong), being a descriptive, observational and comparative analysis rather than proof-of-concept investigation.

We thank the reviewer for their useful comments.

Reviewer 3 Report

Technically, the manuscript is well done and the study clearly shows the benefits of immobilizing RGD peptides to synthetic grafts to improve retention and function of endothelium. 

The approach described by the authors is not new and many of the cited studies are 10-15 years old.  In the Discussion, the authors should discuss some of the issues associated with making this approach a viable alternative to improve the patency of small diameter vascular grafts, how this work addresses those issues and remaining challenges.

Author Response

Technically, the manuscript is well done and the study clearly shows the benefits of immobilizing RGD peptides to synthetic grafts to improve retention and function of endothelium.

The approach described by the authors is not new and many of the cited studies are 10-15 years old.  In the Discussion, the authors should discuss some of the issues associated with making this approach a viable alternative to improve the patency of small diameter vascular grafts, how this work addresses those issues and remaining challenges.

We have revised the Discussion section to address your comment. We have added the following paragraph:

The major challenge in developing synthetic small-diameter vascular grafts is to mimic the properties of the native vessel. An ideal material for such grafts should be remodelable by host cells and its degradation products must be easily metabolized and excreted by the organism. Modification of prostheses with bioactive molecules needs to address multiple issues, including control of platelet adhesion, coagulation cascade and thrombus formation, whilst promoting endothelialization. Peptides with different RGD sequence as well as their linkers are completely unstudied in terms of vascular graft modification. The results presented in our study provide an insight on the functionality of small-diameter vascular prostheses in different modes of RGD modification. Further in-depth trials are required to validate our results in patients requiring arterial replacement.

We thank the reviewer for their useful comments.